# α-Synuclein and Mechanisms of Epigenetic Regulation

**DOI:** 10.3390/brainsci13010150

**Published:** 2023-01-15

**Authors:** Andrei Surguchov

**Affiliations:** Department of Neurology, Kansas University Medical Center, 3901 Rainbow Boulevard, Kansas City, KS 66160, USA; asurguchov@kumc.edu

**Keywords:** α-synuclein, epigenetic, DNA-binding proteins, DNA methyltransferase, CpG methylation, chromatin remodeling, microRNAs, histone acetylation, BAP1

## Abstract

Synucleinopathies are a group of neurodegenerative diseases with common pathological lesions associated with the excessive accumulation and abnormal intracellular deposition of toxic species of α-synuclein. The shared clinical features are chronic progressive decline of motor, cognitive, and behavioral functions. These disorders include Parkinson’s disease, dementia with Lewy body, and multiple system atrophy. Vigorous research in the mechanisms of pathology of these illnesses is currently under way to find disease-modifying treatment and molecular markers for early diagnosis. α-Synuclein is a prone-to-aggregate, small amyloidogenic protein with multiple roles in synaptic vesicle trafficking, neurotransmitter release, and intracellular signaling events. Its expression is controlled by several mechanisms, one of which is epigenetic regulation. When transmitted to the nucleus, α-synuclein binds to DNA and histones and participates in epigenetic regulatory functions controlling specific gene transcription. Here, we discuss the various aspects of α-synuclein involvement in epigenetic regulation in health and diseases.

## 1. Introduction

Epigenetics deals with the changes in the expression of genes caused by various stimuli that are not attributed to alterations in the nucleotide sequence of DNA. Epigenetic changes include chemical modifications to DNA, as well as to the proteins bound to DNA. The instruments involved in this regulation consist of DNA methylation (DNAm), chemical modifications of histones, remodeling of chromatin, and non-coding RNAs expression [1,2,3,4,5,6]. These mechanisms play a vital role in various biological processes, including development, normal functions of a healthy organism, and pathology through fine-tuning and coordination of spatial and temporal gene regulation. 

Conrad Waddington [7] proposed the term ‘Epigenetic’ to define the unclear and ambiguous association between genetic information and phenotypic traits. The term was proposed to explain why alterations in gene expression were not always originating from changes in the genetic sequence.

Later, with the accumulation of new data and deeper knowledge in genetics, molecular biology, and biochemistry, aiding in the understanding of the subject, the designation of epigenetics developed and one of the contemporary characterizations may be as follows: “the investigation of stable and transmissible non-genetic effect on expression of genes or cell phenotypic features without affecting the DNA sequence”. Such reversible alterations can regulate gene expression, permitting changes within various cellular phenotypes. The collection of all epigenetic changes in a genome is called an epigenome [5]. Epigenetics became the focus of attention in synucleinopathies because Genome-Wide Association Studies (GWAS) of these diseases have identified plenty of variants associated with disease phenotypes. However, closer analysis demonstrates that most of these variants do not change DNA coding sequences, pointing to an underlying epigenetic mechanism. This makes it a big challenge to identify their function and role in pathology. Recent studies confirm that epigenetic processes play a key, but not completely understood role in synucleinopathies. 

Despite an enormous increase in emerging experimental data about α-synuclein, its normal functions, mechanism of regulation, and crosstalk among pathological events are not completely clear. This deficiency of knowledge creates a hurdle for the search of effective treatment strategies and diagnostic procedures.

## 2. Epigenetic Regulation of α-Synuclein 

DNAm, which usually includes the addition of a methyl group to one of the carbons of the pyrimidine ring of cytosine in the CpG dinucleotide, has been broadly characterized in human individuals. In addition to its plasticity towards genes and the environment, DNAm also has a plausible application as a biomarker because of the similarity of DNAm patterns between the blood and brain [8].

α-Synuclein expression is regulated by multiple mechanisms, one of which is the methylation of intron 1 region [SNCA(i1)] in α-synuclein gene located upstream of the initiation codon ATG, between exon 1 and exon 2. This region contains CpG islands which serve as binding sites for several transcription factors. Binding of transcription factors to these binding sites regulates α-synuclein expression [2,9]. Hypomethylation in this region increases α-synuclein expression which may cause its accumulation and ultimately lead to pathology [10].

Daniele et al. (2018) [11] demonstrated that the SNCA(i1) methylation level was augmented with the age of an individual, reducing α-synuclein concentration in the blood. Physical activity possessed a similar effect, decreasing both the oligomeric and total α-synuclein levels. The authors concluded that DNAm of SNCA(i1) was associated with age, and total α-synuclein concentrations in erythrocytes were decreased with age. DNA methyltransferase (DNMT) content in this study was in good correlation with oligomeric and total α-synuclein, while total α-synuclein concentration was inversely related to the methylation status of SNCA(i1). 

α-Synuclein can retain DNMT1 in the cytoplasmic fraction of neurons (Figure 1) affecting DNAm. As a result, reduced methylation was detected in SNCA(i1) [12].

Several factors, including variations in the α-synuclein gene, physical activity of an individual, exposure to pesticides, etc. may impact the epigenome, particularly at the level of CpG methylation [5,11,12]. These factors are key contributors to the pathology of synucleinopathies. The results of many recent studies demonstrate the importance of the interaction between genetic and environmental contributions to the disease risk, presumably acting across multiple omics layers [5,13].

The methylation level of SNCA(i1) is different in Parkinson’s disease patients and control individuals in blood, leukocytes, and mononuclear cells. The results of recent investigations demonstrate that the level of methylation in patients is considerately lower than in a control group. A similar conclusion was made when brain tissue methylation was analyzed [14,15,16]. Thus, complex interactions between genotype, environment, lifestyle, and developmental stage affect and regulate epigenetic modifications [17,18].

## 3. α-Synuclein as a Genetic Modulator

### 3.1. Nuclear Localization of α-Synuclein and DNA Binding

After finding α-synuclein in the nucleus and presynaptic nerve terminals [19], the association of neuritic and nuclear α-synuclein, and the functional activity and interacting partners of α-synuclein in subcellular compartments, have been the focus of attention of many investigations. However, the biology of nuclear α-synuclein is still not completely apprehended.

The interactions of both wild-type and mutant forms of α-synuclein with dsDNA were described for the first time by Cherny et al. (2004) [20], and later confirmed by several studies [21,22,23]. α-Synuclein-DNA binding alters the properties of both interacting molecules. α-Synuclein–DNA interaction changes DNA bending properties and increases aggregation of the protein, with a concomitant fibril formation. Although, at present, a clear answer to the meaning of this interaction is under discussion, an interesting hypothesis is that α-synuclein binds to histone-free segments of DNA which are transcriptionally active, and that this interaction causes a transcription activity alteration [20]. Binding of α-synuclein to specific nucleotide sequences in DNA has been later repeatedly confirmed [23,24,25,26].

α-Synuclein binding to DNA modulates several DNA- and RNA-dependent activities, including transcription, rRNA and mRNA metabolism [25,27,28], and DNA double-strand break repair [24]. Moreover, α-synuclein induces DNA stretches and increases its stiffness [29]. However, probably the most exciting and stimulating were the studies investigating the involvement of α-synuclein in epigenetic mechanisms, for example its role in the regulation of histone modification [30].

The biochemical functions of nuclear α-synuclein demonstrate that it binds both to DNA and histones. This interaction is physiologically important because it controls DNA repair and transcriptional regulation [24,31]. Interestingly, during neuronal differentiation, distinct α-synuclein species translocate from the nucleus to neuronal processes. α-Synuclein nuclear–somatic–neuritic shuttling is a dynamic process during neuronal differentiation. Nuclear α-synuclein induces noticeable transcriptional deregulation, for example, the downregulation of key cell cycle-related genes. This effect on transcription is related to the reduced α-synuclein binding to DNA. Increased nuclear presence of high-molecular weight α-synuclein regulates specific gene expression. The effect of nuclear α-synuclein on gene expression and cytotoxicity is modulated by phosphorylation on serine 129 (Ser^129^, Figure 2) [31]. Thus, the activity of nuclear α-synuclein is modulated by the level of its aggregation and post-translational modifications. 

Pinho and co-authors [31] investigated the activity of aggregated and phosphorylated α-synuclein in the nucleus, comparing the results of studies in postmortem brain samples and in different cell models. The authors found that phosphorylation of α-synuclein modulated both its nuclear localization and its role as a regulator of transcription. The results of the epitope dot blot analysis suggest that different conformational states of nuclear and cytosolic α-synuclein with exposed different epitopes determine its interaction with other compounds in the cytoplasm and nucleus [31].

α-Synuclein also interacts with a protein BAP1 (BRCA1 associated protein 1) possessing ubiquitin C-terminal hydrolase activity (UCH) [38]. BAP1, possessing deubiquitinase activity (DUBs), is required for the reverse reaction of ubiquitination, acting as a major regulator of ubiquitin signaling processes. BAP1 deubiquitinating enzymatic activity is an important modulator of gene expression. The interaction of α-synuclein with BAP1 and other DUBs is a prognostic indicator of unfavorable outcome in several types of cancer, and may also affect neurodegenerative processes [38]. Further studies are necessary to better understand the regulatory role of α-synuclein-BAP1 interaction and its effect on specific gene regulation. 

### 3.2. Nuclear α-Synuclein Regulates Histone Modifications

α-Synuclein interactions with nuclear components are not restricted to DNA. α-Synuclein binds to histones, and its fibrillation is accelerated by histone H1 and other core histones. Co-localization of α-synuclein with histone H1 and H3 and formation of a complex in the nucleus is proven by immunohistochemistry data using neuronal nuclei marker NeuN [39]. The condensation of histones is basically responsible for the organization of euchromatin and heterochromatin, the first being active and the latter transcriptionally repressed. The activity of proteins performing transcription is regulated by various biochemical modifications, i.e., methylation, phosphorylation, and acetylation, on DNA itself or on histones. Post-translational modifications of histone tails may change the protein surface charge, altering its affinity for DNA and for other histones. This, in turn, allows or restricts the entrance of transcription factors and other transcription-associated proteins. Epigenetic “writers”—the proteins responsible for writing these marks—are able to add modifications to histones or DNA. This group includes DNMTs, HAT, and histone methyltransferases (HMTs). These proteins function on the chromatin and make dynamic modifications in response to the environmental factors. Corresponding “erasers”, e.g., histone deacetylases (HDACs), remove alterations to histones or DNA that were introduced by epigenetic writers. 

α-Synuclein may interact with epigenetic writers. For example, expressing α-synuclein raises histone-H3 lysine-9 (H3K9) methylation [40]. Overexpressed α-synuclein can retain DNMT1 in the neuronal cytoplasm, changing DNAm (Figure 1). Moreover, inducible α-synuclein expression increases the level of mono- and dimethylation. These changes may cause the release of synaptic vesicles, contributing to the synaptic dysfunction that occurs in PD [40].

Another important mechanism of α-synuclein’s regulatory effect on transcription is via its interaction with epigenetic erasers. Extensive evidence has demonstrated that α-synuclein restricts HDACs in the cytoplasm, constraining its normal function. Experiments with mutant forms of α-synuclein, e.g., pA^30^P and pA^53^T, showed that it bound to histones, which in turn decreased histone acetylation (Figure 1). Thus, HDACs inhibition protects against α-synuclein’s toxic effect, as found in several model systems such as SH-SY5Y cells and transgenic flies [30]. 

α-Synuclein interacts with chief epigenetic eraser HDAC4, abundant in neurons. It forms a part of the HDACs class IIa that can shuttle between the nucleus and cytoplasm [41]. Exposition to MPTP (1-methyl-4-phenyl-1,2,3,6-tetrahydropyridine) causes accumulation of HDAC4 in the nucleus. Moreover, nuclear HDAC4 mediates the cell death in A^53^T cells by inhibiting CREB (cAMP response element binding protein) and MEF2A (myocyte enhancer factor 2A) [42]. 

Finally, it has been reported on H4 cells that the pS^129^-α-synuclein in the nucleus downregulates important cell-cycle genes, e.g., G2/mitotic-specific cyclin-B1 (CCNB1)—a protein involved in mitosis and a transcription factor modulating the cell cycle (E2F8) [31].

Furthermore, Schaser et al. [24] described a new α-synuclein role in the nucleus. A team of researchers demonstrated that α-synuclein phosphorylated on serine-129 (pS^129^) rapidly translocated to laser-induced DNA damage sites in the nucleus of in vivo mouse brain cells, as well as in a mouse primary cortical neuron system. This α-synuclein recruitment had a conceivable role in the DNA break repair [24]. This pathological form is present in the nucleus of neurons in brain areas of mice with impaired cognitive behavioral phenotypes. α-Synuclein’s role in the interaction and modification of histones is essential because histones are key regulators of transcription, and α-synuclein binding alters histones regulatory functions.

### 3.3. α-Synuclein Mediated Histone Modification Regulates Transcription

α-Synuclein’s interaction with histones and their enzymatic modification alter histone’s properties, being important mechanisms of transcription regulation. For example, addition of an acetyl group to a lysine on a histone (Ac, Figure 1) counteracts the positive charge on the residue, preventing the interaction between the DNA and histone, and activating transcription. This reaction is catalyzed by histone acetyltransferases (HATs) (Figure 1). In a reverse reaction, an acetyl group may be removed by HDACs. 

HATs, which are sometimes called KATs, catalyze acetyl group transfer from acetyl CoA to ε-N-acetyl lysine on histone proteins (Figure 1). On the other hand, the reaction of histone deacetylation catalyzed by HDACs induces the formation of a specific closed chromatin conformation. Thus, HDACs are associated with inhibition of transcription.

Histone methylation. Two enzymes control histone methylation, histone methyltransferases (HMTs) and histone demethylases (HDMs). They both transfer methyl groups on the lysine or arginine residues of histone proteins. Lysine residues can accept one, two, or three methyl groups, while arginine residues can accept one or two methyl groups in asymmetrical or symmetrical positions. The methylation of histones specifically regulates transcription, being either an activator or inhibitor of transcription in specific sites, depending on the methylated residue. For example, methylation of histone H3 on lysine 4, arginine 17, lysine 36, or lysine 79 activates transcription. On the other hand, methylation of histone H4 on lysine 20, or histone H3 at lysine 9 and lysine 27 is often associated with transcriptional repression. Histone lysine methyltransferases (KMTs) catalyze histone lysine methylation, whereas histone lysine demethylases (KDMs) catalyze the removal of methyl groups [1,4,43]. α-Synuclein interacts with several epigenetic writers. In transgenic flies expressing human wild-type α-synuclein, histone-H3 lysine9 (H^3^K^9^) methylation is significantly increased [40]. Moreover, overexpression of α-synuclein retains DNA methyltransferase 1 in the neuron’s cytoplasm, thus compromising DNAm (Figure 1) [12]. 

Interestingly, SH-SY5Y cells with inducible expression of human wild-type α-synuclein have advanced levels of H3K9 mono- and dimethylation (Figure 1) leading to synaptic dysfunction, which often happens in Parkinson’s disease [40].

In addition to epigenetic writers, α-synuclein can also interact with epigenetic erasers. α-Synuclein may cause restriction and maintenance of HDACs in the cytoplasm, suppressing their usual function. Human mutant forms of α-synuclein, i.e., p.A^30^P and p.A^53^T bind to histones, lowering histone acetylation, which is key for transcriptional activation (Figure 1). Accordingly, inhibiting HDACs decreases the level of α-synuclein toxicity, as proved by the studies using several model systems [30].

α-Synuclein also modulates HDAC4 activity, which is an important epigenetic eraser. It is expressed in neuronal cells and forms part of the HDACs class IIa that can shuttle between the nucleus and cytoplasm [41]. Indeed, exposing mice with expression of p.A^53^T-mutant α-synuclein or PC-12 cells to a sub-toxic concentration of MPTP causes accumulation of HDAC4 in the nucleus. HDAC4 accumulation in the nucleus mediates cell death in p.A^53^T cells by inhibiting the transcriptional activity of CREB and MEF2A (myocyte enhancer factor 2A) [42].

Important experimental results have been obtained using H4 cells, demonstrating that the α-synuclein phosphorylated on serine 129 has higher affinity for the nucleus, downregulating genes CCNB1 and E2F8 that control the cell cycle [31]. These results indicate a probable effect of mutant α-synuclein on progression between phases of the cell-cycle. Schaser et al. [24] demonstrated that pS^129^-α-synuclein was quickly translocated to DNA damage sites in the nucleus of a mouse primary cortical neuron system, as well as in in vivo mouse brain cells. These results point to a plausible α-synuclein role in double-strand break repair [24]. This α-synuclein form has been found within the neuronal nuclei in several brain parts, e.g., the hippocampus from aged p.A^30^P α-synuclein mice with impaired cognitive behavioral phenotypes, and basolateral amygdala and cortex. On the other hand, these alterations were not detected in young p.A^30^P α-synuclein mice [44]. Thus, phosphorylation of α-synuclein might also be linked to an aging process.

The biochemical function of nuclear α-synuclein is not completely clear, and the investigation of its interactions with nuclear regulatory elements has re-appeared after the current advances in epigenetic mechanisms, mostly the consequences involving Parkinson’s disease. Recent results show that α-synuclein affects mechanisms of transcriptional regulation by interacting with both epigenetic writers and erasers. These interactions result in upregulation of its own transcription, thus leading to its accumulation and aggregation within the cell. This sequentially disturbs the normal function of cellular organelles compromising cell viability. Thus, discovering mechanisms by which α-synuclein transcription may be downregulated to avoid its aggregation may prevent, or at least reduce, the probability of neuronal death.

### 3.4. α-Synuclein and Chromatin Remodeling

A variety of cellular stresses and protein–protein interactions induce α-synuclein’s translocation to the nucleus. Nuclear α-synuclein interacts with DNA, histones, affects gene expression, and regulates chromatin remodeling [45,46,47,48]. The mechanism underlying α-synuclein crosstalk in chromatin remodeling in synucleinopathies, including Parkinson’s disease, is an exciting area of investigation. Lee et al. [48] studied the role of nuclear α-synuclein in neurotoxicity and chromatin remodeling. The authors examined transcriptional adapter 2-alpha (TADA2a) as an α-synuclein binding protein. TADA2a is a constituent of the p300 (HAT)/CBP-associated factor involved in histone H3/H4 acetylation.

Sun et al. (2020) [49] applied analysis of frequent gene co-expression using substantia nigra-specific microarray datasets. The authors combined small-scale genetic screening in model organisms with bioinformatics analysis of large-scale human transcriptomic data. As a model, they used Drosophila with α-synuclein-mediated changes mimicking Parkinson’s disease. This approach identified new disease-related genes including SMARCA4—a chromatin-remodeling factor and a BLVRA—biliverdin reductase. SMARCA4 (transcription activator BRG1) is similar to Drosophila brahma protein. Members of this family possess ATPase and helicase activities, and regulate the transcription of specific genes by changing the structure of chromatin in the vicinity of these genes. BLVRA is involved in the biliverdin/bilirubin redox cycle. A role of these new proteins was further studied in a Drosophila model of Parkinson’s disease. Investigations conducted in this model demonstrated that SMARCA4 inhibition prevented an aging-dependent dopaminergic degeneration. Moreover, reduction in SMARCA4 expression in the dopaminergic neurons averted the decrease in life span related to LRRK2 and α-synuclein. These results unveil a new role of epigenetic regulators, and point to a novel epigenetic target in addition to HDACs and DNMTs for the therapeutic interventions of Parkinson’s disease and other neurodegenerative disorders. The authors assume that these findings identified multiple therapeutic targets. The manipulation with these targets or entry points might reverse DA degeneration, postpone the beginning of organ aging, and extend the life span.

## 4. Non-Coding RNAs Regulate α-Synuclein Expression

The results of recent investigations have indicated that non-coding RNAs (ncRNAs) regulate α-synuclein expression. Since α-synuclein upregulation is a critical step that promotes its aggregation in Lewy bodies, ncRNAs are considered to be important players in synucleinopathies [50]. NcRNAs are a heterogeneous group of RNAs that play a significant but not completely understood role in epigenetic regulation.

NcRNAs may be subdivided into two groups: housekeeping ncRNAs and regulatory ncRNAs. The regulatory ncRNAs belong mostly to two types classified on the basis of their base-pair length [3,51]: long ncRNAs (lncRNAs) and small ncRNAs (i.e., siRNAs, miRNAs, and piRNAs). NcRNAs and microRNAs (miRNAs) are regulatory molecules involved in the regulation of gene expression. LncRNAs have a length of more than 200 nucleotides. This type of ncRNAs regulates the expression of multiple genes in an epigenetic, transcriptional, or post-transcriptional manner [52], including genes implicated in synucleinopathies [53]. 

Liu et al. [54] demonstrated that lncRNA NEAT1 promoted α-synuclein transcription, and enhanced the Bax/Bcl ratio and activity of caspase 3 in Parkinson’s disease. These results point to the role of NEAT1 in apoptosis associated with α-synuclein.

In another study, Lin et al. [55] revealed a role of miR-519-3p and LncRNA-T199678 in α-synuclein pathology related to Parkinson’s disease. Overexpression of lncRNA-T199678 decreases α-synuclein-induced neurotoxicity interacting with intracellular oxidative stress. These results point to the existence of α-synuclein/lncRNA T199678/miR-519-3p pathway related to intracellular oxidative stress, which plays an important regulatory role in PD-related α-synuclein pathology [55,56]. In another study, Wang et al. [57] proposed an α-synuclein/miR-101–3p/lncRNA-T199678 pathway associated with oxidative stress that plays a key role in the pathogenesis of Parkinson’s disease [57]. Another type of ncRNA—MALAT1 (or NEAT2)—a large RNA expressed in the nucleus, is related to α-synuclein aggregation [58]. MALAT1 binds to α-synuclein enhancing its stability; this interaction leads to a higher abundance of α-synuclein. Accumulating evidence shows that lncRNAs can function as miRNA decoys or sponges to compete for miRNA binding to protein-coding transcripts. Sun et al. (2022) [59] investigated the function of lncRNA HOX transcript antisense RNA (HOTAIR) in Parkinson’s disease. The authors revealed that LncRNA HOTAIR promoted α-synuclein aggregation and apoptosis by regulating miR-221-3p. HOTAIR sponged miR-221-3p which directly targeted α-synuclein and positively regulated its expression. The results point to the miR-221-3p/α-synuclein axis as a potential therapeutic target in Parkinson’s disease, and to HOTAIR as an efficient miRNA sponge. The protective effect of MiR-221 against Parkinson’s disease was described in several other studies using various mice models [60,61]. The effect of the regulatory role of lncRNAs may be mediated by other types of miRNAs, i.e., miR-34b-5p and miR-133b [62,63]. 

## 5. Epigenetic-Based Approaches for Treatment and Diagnosis of Neurodegenerative Diseases 

Epigenetic modulators responsible for gene expression regulation and modulation of the chromatin structure may be potentially used for the diagnosis and treatment of human neurodegenerative diseases. Examples of such modulators are inhibitors affecting HDAC family members [15,64]. For instance, suberoylanilide hydroxamic acid (SAHA) is an FDA-approved drug for the treatment of subcutaneous T cell lymphoma, which can be also used for the treatment of neurological diseases. SAHA is able to cross the blood–brain barrier and can improve memory function [64,65,66,67]. Treatment with HDAC inhibitors triggered the increase in the expression of a set of genes controlling neuroplasticity, accompanied by advanced histone acetylation in the respective promoter region. As a result, even remote memories were persistently attenuated by SAHA treatment [68].

Another type of epigenetic modulator which can be used for synucleinopathies treatment is microRNAs. In particular, several microRNAs possessed a therapeutic potential and provided neuroprotection by targeting Parkinson’s-associated genes.

Roser et al. [69] described the neuroprotective effect of miR-182-5p and miR-183-5p, both of which downregulated GDNF (glial cell-derived neurotrophic factor) and protected dopaminergic neurons against neuronal damage.

In another study, Wang and Deng [70] defined a role of miR-7 and lncSNHG1, miR-223-3p, and lncGAS5 which interact together in the pathogenesis of Parkinson’s disease. These RNAs act through the mechanism of NLRP3 activation, and can be used in the future for early diagnosis and probably for the development of new treatment options. Antagonism of CDR1AS and circSNCA with miR-7 has been described, preventing dopaminergic neuron damage and neuroinflammation. 

Pharmacologically targeting of one of the HDACs deacetylases by specific inhibitors may be of potential therapeutic benefit. HDACs inhibitors, for example, valproic acid, phenylbutyrate, sodium butyrate trichostatin A (TSA), and SAHA can alleviate endoplasmic reticulum stress, and ameliorate motor deficit by behaving as a potential therapeutic agent. They may be more effective in a combinatorial setting with other agents [71,72]. 

Examination of differentially methylated replicating loci may also serve as a tool for the diagnosis of synucleinopathies. In particular, this approach may be applied for Parkinson’s disease and dementia with Lewy bodies—related progressive disorders neuropathologically associated with the accumulation of intraneuronal aggregates of misfolded α-synuclein. 

Pihlstrøm and co-authors [73] performed an epigenome-wide association study (EWAS) of postmortem frontal cortex samples to explore the role of DNA methylation changes in the pathogenesis of Parkinson’s disease and dementia with Lewy bodies. This study revealed that new differentially methylated replicating loci provided possible evidence for a locus within the chromosomal region affected by the Parkinson’s-associated deletion of 22q11.2 chromosome segment. These findings identified novel disease pathways in these two forms of synucleinopathies, and suggested an approach for differential diagnosis of two close forms of these disorders [73]. 

## 6. Conclusions and Further Developments 

There is no disease-modifying treatment for synucleinopathies and no reliable biomarkers to identify early symptoms of these diseases. Since both of these concerns still remain unsolved, the vigorous and urgent search for new potential targets is currently underway. During the last few years, a growing number of studies are focused on the role of epigenetic factors in the regulation of gene expression in health and diseases. The epigenome consists of numerous factors, other than the DNA sequence alterations, that affect transcription, chromatin interactions, and other mechanisms.

It is currently common knowledge that epigenetic mechanisms play a key role in the pathogenesis of synucleinopathies. Together with the contribution of genetic alterations, epigenetic changes are also implicated in the advancement of these disorders. Thus, epigenetic-based approaches for the treatment of neurodegenerative diseases are currently required due to their high specificity towards the target molecule, and the distinctive nature of the alterations to be reversible. Further studies in this direction may offer new prospects in better understanding the pathogenesis of the synucleinopathies, and help to identify an enhanced therapeutic approach [74]. Recent findings demonstrate that epigenetic mechanisms may act via complex multimolecular interactions between lncRNAs, miRs and BAP1, which act through the miR-150-5p/BAP1 axis [75]. Most of these alterations are associated with ageing. Therefore, the new results of epigenetic studies will help to define the therapeutic interventions, and also clarify the mechanisms of aging as a risk factor for the occurrence of synucleinopathies. More research is also needed for deciphering the dynamic association of DNA/RNA methylation, chromatin remodeling, histone modification, non-coding RNAs, etc., with environmental factors. 

## Figures and Tables

**Figure 1 brainsci-13-00150-f001:**
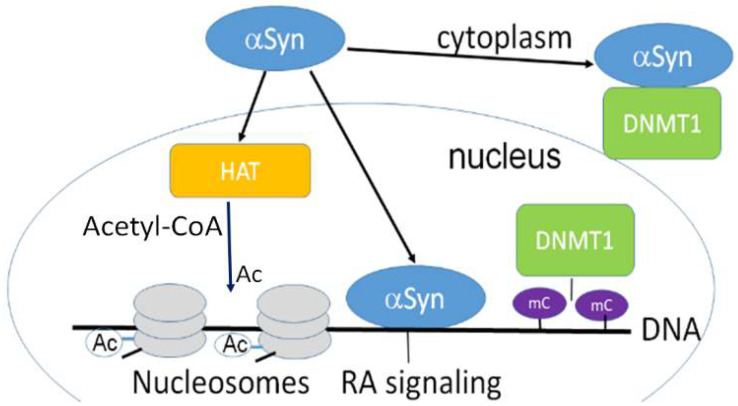
α-Synuclein is confined both in the nucleus and cytoplasm and is involved in epigenetic regulation. Nuclear α-synuclein binding to DNA regulates specific gene transcription, reducing histone acetyltransferase p300 (HAT) activity. α-Synuclein is able to sequester DNMT1 from the nucleus to the cytoplasm [5]. HAT acetylates conserved lysine residues on histones. This reaction includes the transfer of an acetyl group (Ac) from acetyl-CoA to form ε-N-acetyl-lysine. Histone acetylation switches genes on and off, acting as a regulator of gene expression. DNA is wrapped around histones forming nucleosomes (gray disks). Nuclear α -synuclein also participates in the retinoic acid (RA) regulation of the signaling pathway. mC-CpG methylation.

**Figure 2 brainsci-13-00150-f002:**
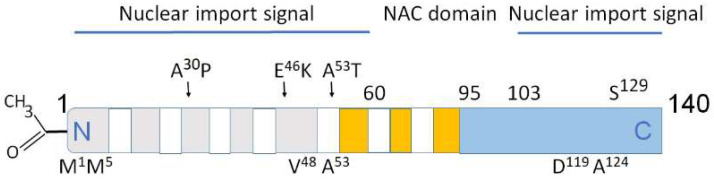
Structural domains in α-synuclein involved in aggregation, metal binding and nuclear transport. Above the rectangle: A^30^P, E^46^K, A^53^T—some of mutations causing PD. Below the rectangle: M^1^, M^5^ V^48^, A^53^—high affinity Cu binding sites [32,33]. The predominantly disordered C-terminal domain modulates interactions of pro-oxidant metals associated with PD pathology. D^119^, A^124^—divalent metals Fe, Cu, Co, and Mn binding sites. Amino acids 1–60 and 103–140 are important for nuclear transport (nuclear import signals). The core sections of the seven amino-terminal repeats (KTKEGV motive) are shown as white bars. These motives are located between amino acids 7–87. Five positively charged regions are light gray, three negatively charged hydrophobic regions are orange, C-terminal region is light blue. Non-amyloid-β component (NAC domain) constituting amino acids 61–95 is critical for aggregation [33,34,35]. N-terminal acetylation (left) slows down α-synuclein aggregation and changes the morphology of the aggregates [36,37].

## Data Availability

Not applicable.

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
