# Peer review of "α-Synuclein and Mechanisms of Epigenetic Regulation"

_brainsci, 2023, doi:10.3390/brainsci13010150_

Round 1

Reviewer 1 Report

In this review manuscript, the author presents an elaborate summary of a-Synuclein and the mechanisms involved in its epigenetic regulation. As synucleinopathies are among the most common neurodegenerative diseases, they form an important field and warrant further development. This comprehensive review should be helpful for researchers to understand the importance and the latest progression of epigenetic regulation in synucleinopathies.

In general, the review is well-written and comprehensive. However, it is packed with a lot of scattered information and thus goes short on clear readability.

I have some suggestions that are given below –

1. The entire document lacks a clear flow. Different modules seem to be independently written and not merged properly.

2. Making a clear distinction between synuclein as the main driver of changes, and other factors that drive changes in synuclein would greatly increase its readability. Right now, it is not always clear what causes changes in what and how synuclein is central to this. I propose to first write epigenetic regulation of a-synuclein, and then write a-synuclein as a genetic modulator. Or focus on the latter only.

3. I suggest adding an all-encompassing figure showing all possible mechanisms that a-synuclein carries out as an epigenetic modulator, and a table recapitulating all these details would also be helpful for clarity.

4. line 86 – provide a reference.

5. lines 95-100 – they seem introductory and abrupt, and may fit better after line 52.

6. line 169- correct the spelling of writers

7. line 170 – DNMTs and HAT are already introduced. Using only acronyms is okay.

Author Response

Responses to Reviewer 1

1 The entire document lacks a clear flow. Different modules seem to be independently written and not merged properly.

Response: Thank you for this comment. We corrected the text making better connections between different modules.  We also changed some sub-headings to make better bridges between  modules.

2 I propose to first write epigenetic regulation of a-synuclein, and then write a-synuclein as a genetic modulator. Or focus on the latter only.

Response: Thank you for this suggestion. We did as recommended: the title of the second module is now changed as “Epigenetic regulation of a-synuclein” and subsequent module 3 is renamed as “a-Synuclein as a genetic modulator” with subsequent use of sun-headings and corresponding changes in enumeration (3a, 3b etc.)  We believe that these alterations improved the readability of the manuscript.

 3 I suggest adding an all-encompassing figure showing all possible mechanisms that a-synuclein carries out as an epigenetic modulator, and a table recapitulating all these details would also be helpful for clarity.

Response: Thank you for this suggestion. We believe that for the current mini review the information in Figures 1 and 2 is sufficient for understanding. We plan to write a more comprehensive review in which we will include additional figures and Tables.

4 line 86 – provide a reference.

Response: we added citations to references 11 and 12.

 5 lines 95-100 – they seem introductory and abrupt, and may fit better after line 52.

Response: We agreed and deleted lines 95-99

 6 line 169- correct the spelling of writers

Response: we corrected the spelling of “writers” (in a new version this line became 171).

 7 line 170 – DNMTs and HAT are already introduced. Using only acronyms is okay.

Response: Thank you. We corrected it and used acronyms.

We would like to thank reviewer 2 for valuable and helpful suggestions and criticism

Reviewer 2 Report

The manuscript under review is an overview on the topic of a potential role of α-Synuclein in epigenetic modulation in health and disease based on current research. The authors also suggest several therapeutic strategies including modulation of the chromatin structure that are indicated to have the potential to slow or impede neurodegeneration. However, the author does not take into account findings from other neurodegenerative diseases, such as Alzheimer's or Huntington's, that may inform the mechanisms underlying Synucleinopathies.

1.     Page 1, line 41-43 needs reference.

2.     The authors must explain the abbreviations when used for the first time like BAP1 in line 152.

3.     BAP1 has been used as a keyword for the manuscript yet not much information has been given by the authors on the role of BAP1 and its interaction with α-Synuclein.

4.     Under the section, α-Synuclein: nuclear localization and DNA binding, line 107-110: authors do not clarify if the α-Synuclein that interacts with the DNA and histone and alters transcription is the wild type α-Synuclein or its mutant form. Also, what role does α-Synuclein interaction with DNA and histones play in general?

5.     Authors must add an elaborative figure showing the interaction with all the epigenetic mechanisms.

6.     The conclusive section needs to be more focused.

Author Response

Responses to Reviewer 2

1 Page 1, line 41-43 needs reference.

Response: We added reference 5 on line 41.

2 The authors must explain the abbreviations when used for the first time like BAP1 in line 152.

Response: We added explanation as follows: BAP1 (BRCA1 associated protein 1).

3  BAP1 has been used as a keyword for the manuscript yet not much information has been given by the authors on the role of BAP1 and its interaction with α-Synuclein.

Response: We added the following “BAP1 possessing deubiquitinase activity (DUBs) is required for the reverse reaction of ubiquitination acting as a major regulator of ubiquitin signaling processes.”BAP1 deubiquitinating (DUBs) enzymatic activity is an important modulator of gene expression. The interaction of α-synuclein with BAP1 and other DUBs is a prognostic indicator of unfavorable outcome in several types of cancer and may also affect neurodegenerative processes [38]. Further studies are necessary to better understand the regulatory role of α-synuclein-BAP1 interaction and its effect on specific gene regulation.

We also added the following text in Conclusion:

“Recent findings demonstrate that epigenetic mechanisms may act via a complex multimolecular interactions between lncRNAs, miRs and BAP1 which acts through miR‑150‑5p/BAP1 axis [75].” We also included reference [75] in response to reviewer’s comment.

4 Under the section, α-Synuclein: nuclear localization and DNA binding, line 107-110: authors do not clarify if the α-Synuclein that interacts with the DNA and histone and alters transcription is the wild type α-Synuclein or its mutant form. Also, what role does α-Synuclein interaction with DNA and histones play in general?

Response: as indicated now on line 109: ”both wild-type and mutant forms of α-synuclein with dsDNA”

5  Authors must add an elaborative figure showing the interaction with all the epigenetic mechanisms.

Response: We don’t think we can elaborate a figure showing the interaction with all the epigenetic mechanisms except those presented on Figure 1.

6 The conclusive section needs to be more focused.

Response:  We made some additions to Conclusion, focusing on assumptions and hypotheses.

We would like to thank reviewer 2 for valuable and helpful suggestions and criticism